# Volcanism in slab tear faults is larger than in island-arcs and back-arcs

Luca Cocchi [1], Salvatore Passaro [2], Fabio Caratori Tontini [3] & Guido Ventura [1,2]

Subduction-transform edge propagators are lithospheric tears bounding slabs and back-arc basins. The volcanism at these edges is enigmatic because it is lacking comprehensive geological and geophysical data. Here we present bathymetric, potential-field data, and direct observations of the seafloor on the 90 km long Palinuro volcanic chain overlapping the E-W striking tear of the roll-backing Ionian slab in Southern Tyrrhenian Sea. The volcanic chain includes arc-type central volcanoes and fissural, spreading-type centers emplaced along second-order shears. The volume of the volcanic chain is larger than that of the neighbor island-arc edifices and back-arc spreading center. Such large volume of magma is associated to an upwelling of the isotherms due to mantle melts upraising from the rear of the slab along the tear fault. The subduction-transform edge volcanism focuses localized spreading processes and its magnitude is underestimated. This volcanism characterizes the subduction settings associated to volcanic arcs and back-arc spreading centers.

---

[1] Istituto Nazionale di Geofisica e Vulcanologia, Via di Vigna Murata 605, 00143 Rome, Italy. [2] Istituto per l'Ambiente Marino Costiero, Consiglio Nazionale delle Ricerche, Calata Porta di Massa, 80133 Napoli, Italy. [3] GNS Science, 1 Fairway Drive, Avalon, 5010 PO Box 30-368, Lower Hutt 5040, New Zealand. Correspondence and requests for materials should be addressed to G.V. (email: guido.ventura@ingv.it)

Volcanism of subduction settings concentrates in arcs, island-arcs, and back-arc basins[1, 2]. Island-arcs are commonly characterized by composite stratovolcanoes[3], whereas back-arc volcanoes show a morphology similar to that of oceanic spreading centers and rifts, including small cones, flat-top edifices, fissural vents, and spreading volcanoes[4]. Many subduction zones including Tonga, New Hebrides, Lesser Antilles, South Sandwich, and Ionian Sea-Calabrian Arc are, however, affected by a still poorly known volcanism associated to lithospheric tear faults known as subduction-transform edge propagators (STEP)[5–8]. Volcanoes at STEP are located at the lateral margins of retreating slabs[9] but their morphology, structure, and volume of the erupted products are still unknown. In addition, the relationships among island arc, back-arc, and STEP volcanoes need to be investigated to fully understand the dynamics of active subduction zones, reconstruct past subduction settings, and identify possible plate margins. Here, we present high-resolution bathymetric, magnetic, and gravimetric data on the E-W striking STEP fault system delimiting the northern side of the Ionian Sea-Calabrian Arc subduction (Italy). The Ionian Sea-Calabrian Arc is part of the Africa–Eurasia convergence in the Mediterranean Sea and includes the northwestward subduction of the Ionian lithosphere below the Calabrian Arc[10–13] (Fig. 1a). To the northwest of the Calabrian Arc, the Marsili back-arc basin opened about 2 Ma ago as a consequence of the slab roll back[14]. The basin is characterized by 10–12 km thick oceanic crust[15] and its spreading ridge is the NNW–SSE elongated Marsili Seamount[16]. The back-arc opened with a rate 2.8–3.1 cm/yr and, in the last 1 Ma, suffered a decelerating phase[17, 18]. The Aeolian island-arc volcanoes were emplaced on continental crust[19] in the last 1 Ma (Fig. 1a). Tomographic images[20–22] and petrological data[19, 23, 24] indicate that the asthenospheric mantle refills the back-arc and the easternmost Aeolian volcanoes[25] (Fig. 1b); here, these mantle melts mix with fluids derived from the dehydratation of the sub-vertical Ionian slab[20, 23]. Two main tear faults delimit the western and northern boundaries of the Ionian slab[19, 20, 25]: a NNW–SSE striking, dextral strike-slip fault system (Tindari Letojanni fault, TL fault in Fig. 1a) crossing the central sector of the Aeolian island-arc, and an E-W striking STEP fault system (hereafter Palinuro STEP; Fig. 1a) along which the Palinuro and Glabro seamounts were emplaced between 0.8 and 0.3 Ma[11]. The up to 500 km deep seismicity of the Ionian slab concentrates between the Palinuro STEP and TL fault (Fig. 1a). Focal mechanisms of crustal earthquakes on the Palinuro STEP (Fig. 1a) and morphostructural data on the Palinuro Seamount[26] indicate sinistral strike-slip movements along a main, E-W striking shear zone. The Palinuro STEP is a lithospheric boundary separating a 25–30 km thick continental crust from the 10–12 km thick oceanic crust of the Marsili back-arc[15, 27]. A low Vp zone is found at depth of 50–70 km below the Palinuro STEP and a discontinuity in the subducting slab below the Palinuro and Glabro seamounts occurs at 150 km depth[20, 21] (Fig. 1b); high Vp/Vs sill-like anomalies located in correspondence of the Palinuro STEP and Aeolian volcanoes affect the upper 30 km of the crust[22] (Fig. 1c). A toroidal mantle flow from the rear of the Ionian slab to the Palinuro STEP and Tyrrhenian Sea is derived by geodetic and seismic data[25, 28]. Palinuro Seamount (Fig. 1) is presently affected by shallow seismicity and hydrothermal emissions[29, 30]. The discharged gases show a $^{3}He/He^{4}$ anomaly consistent with a deep magmatic-hydrothermal source[31].

We show that a previously unrecognized, up to 3000 m high and ~90 km long submarine volcanic belt strikes along the Plinuro STEP. We analyze the morphology and structure of the volcanic edifices by modeling magnetic and gravity data, and estimate the volume of the erupted products. The results open new perspectives on the volcanism and geodynamics of back-arc subduction zones.

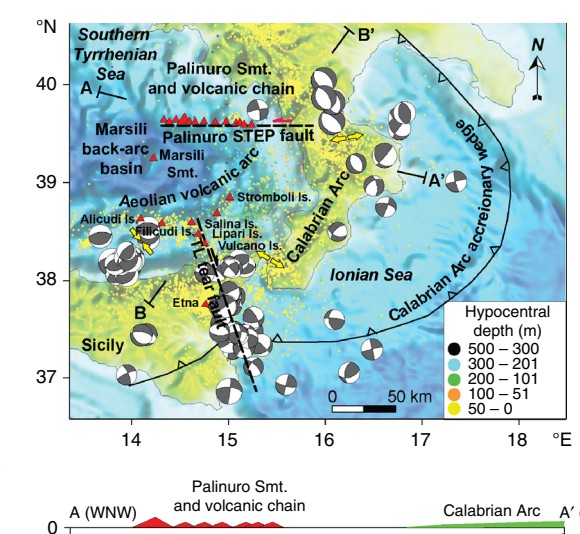

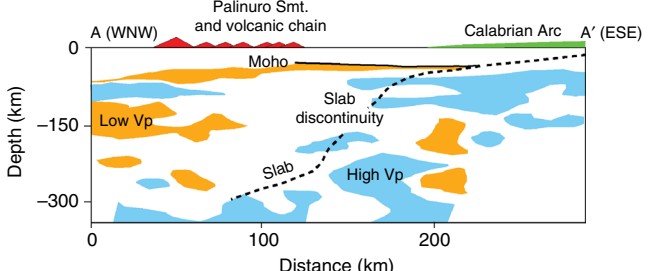

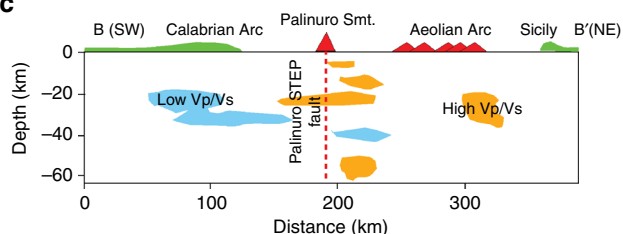

**Fig. 1** Geodynamic setting of the Ionian Sea-Calabrian Arc subduction. **a** The Tindari Letojanni (TL) and Palinuro STEP fault system are from refs. [19, 20, 27]. The 2002–2017 earthquakes are from http://cnt.rm.ingv.it/ and the focal mechanisms of earthquakes with hypocentral depth <30 km and Md > 4 are from http://www.eas.slu.edu/eqc/eqc_mt/MECH.IT/, respectively. **b** Location of the high and low Vp anomalies along the crustal profile A–A′ traced in **a** (data extracted from ref. [21]); the top of the Ionian slab is also reported as dashed line. **c** Location of the high and low Vp/Vs anomalies along the crustal profile B–B′ traced in **a** (data extracted from ref. [22])

## Results

**Morphology.** We collect new bathymetric data and perform a morphometric and morphostructural analysis of the seafloor digital terrain model to reconstruct the physiography of the Palinuro STEP (Methods; Tables 1, 2; Supplementary Figs. 1, 2). So far, only the Palinuro and Glabro seamounts were previously identified as volcanic edifices, whereas our new bathymetric data show that the Palinuro STEP includes a ~90 km long, ~20 km wide strip defined by 15, E-W aligned major volcanoes (Fig. 2a). Hereafter, this volcanic chain is termed STEP volcanic chain. To the west, the STEP volcanic chain southern flanks have a slope of 20–30° and reach the 2700–3000 m deep northern sector of the Marsili Basin, whereas the north flanks have slopes <20° and terminate on a nearly flat surface (slope < 3°) at a depth of about 1600 m (Supplementary Fig. 1). STEP volcanic chain major volcanoes form two main groups separated by a flat area at

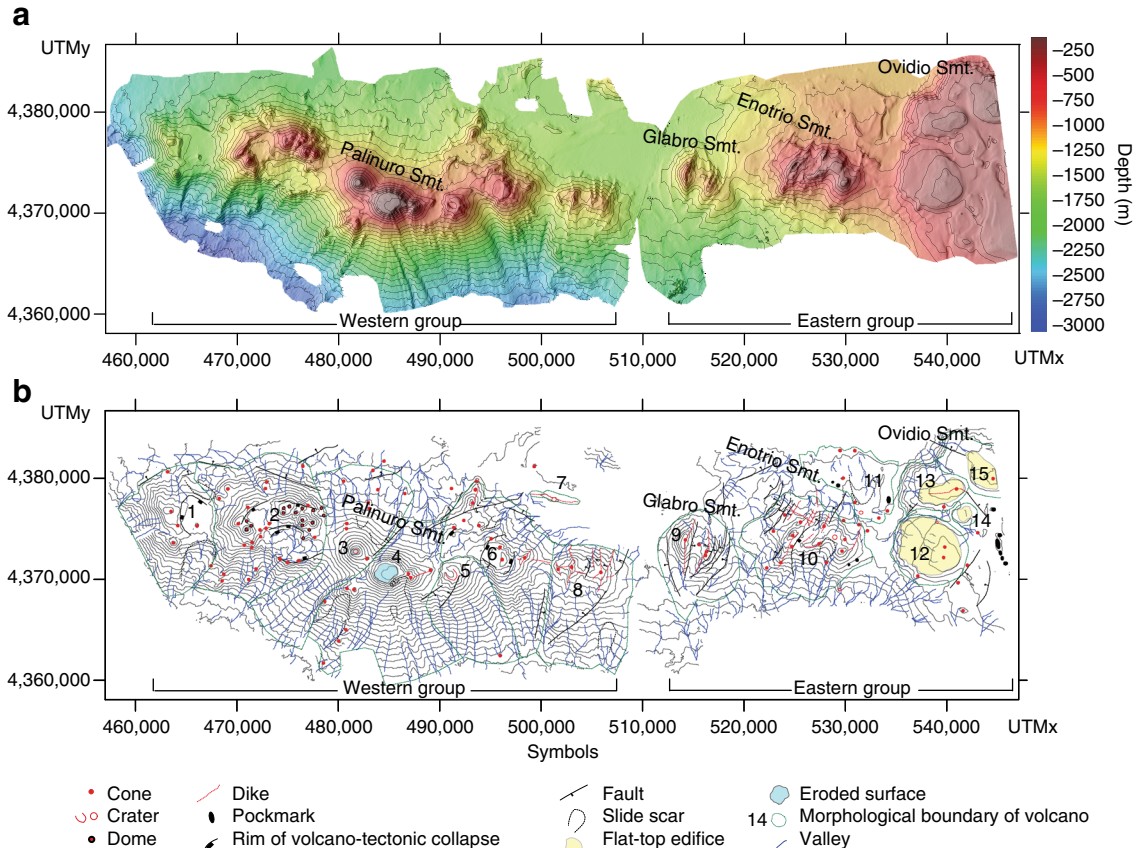

**Fig. 2** Bathymetry and morphostructural setting of the Palinuro STEP volcanic chain. **a** Shaded relief and bathymetry of the 25 m × 25 m STEP Volcanic Chain Digital Terrain Model (WGS85, UTM Zone 33N; see auxiliary material); the contour lines are 100 m spaced. **b** Morphological map of the STEP volcanic chain with the main volcanoes marked by numbers

1500–1600 m depth (Fig. 2a, b). Eighty sub-circular to E–W elongated minor cones with diameter <1 km are also present on the flanks of the major edifices (Fig. 2b). Pockmarks related to gas discharge have been recognized, mainly in the STEP volcanic chain eastern sector (Fig. 2b). The two groups of major volcanoes differ in altitude and slope (Fig. 3a). The western group shows significantly higher slopes and elevations up to 3000 m, whereas the eastern group has lower slopes and elevations not exceeding 1500 m (Figs. 2a, 3a). All the STEP volcanic chain volcanoes are characterized by prevailing north- and south-facing slopes (Fig. 3a; Supplementary Fig. 2). The volcanoes of the eastern group also show second-order west-facing slopes. The STEP volcanic chain western group consists of eight stratovolcanoes, some of which show a sub-circular flat-top surface between 84 and 130 m b.s.l. (Fig. 2b); the westernmost edifices are characterized by a 4 km wide caldera-like depression partly filled by dome-like structures. The flat-top surfaces of the Palinuro Seamount represent terraced landforms formed during low-stand periods in the last 0.4 Ma[32]. The Palinuro steeper flanks are affected by horse-shoe depressions representing slide scars related to seafloor sliding processes (Fig. 2b). The ~8 km long, ~800 m high easternmost edifice of the STEP volcanic chain western group is E–W elongated and dissected by a set of sub-parallel, N–S striking ridges and minor scarps (8 in Fig. 2b). Such morphology is consistent with that of "stellate" volcanoes in which the ridges reflect the emplacement of dikes along rift zones[33]. The morphological features of this volcano suggest the emplacement of a major E–W striking dike from which secondary N–S dikes depart.

The STEP volcanic chain eastern group consists of seven major volcanoes. The ~800 m high Glabro Seamount is dissected by a set of sub-parallel, semi-arcuate and up to 90 m high scarps and ridges defining a N10°E striking spreading zone (Fig. 2a, b). The base of Glabro consists of outward, gently dipping scarps. The morphology of Glabro Seamount suggests spreading processes such as those found in the axial zones of low spreading ridges[34]. The ENE–WSW elongated, 500 m high Enotrio Seamount (summit at 340 m b.s.l.) consists of two major, partly superimposed cones dissected by sub-parallel N10°E to N20°E scarps and ridges. This morphology depicts a rift zone defined by NNE–SSW striking dikes. The top of this volcano is constituted by sheet-like lava flows and dikes (Fig. 4a, b). The easternmost STEP volcanic chain volcanic landforms consist of sub-circular, flat-top edifices with summits between 275 and 315 m b.s.l. and elevations not exceeding 600 m (Fig. 2a, b). Some of these volcanoes show an apical ridge or cone. The summit of the flat-top seamount 12 in Fig. 2b consists of sheet-like lava flows and isolated lava pillows (Fig. 4c, d). The morphology the easternmost STEP flat-top edifices resembles that of mid-ocean ridges[35] and submarine Hawaiian volcanoes, which grow by long-lived steady eruptions with low effusion rates[36].

Up to 10 km long and 90 m high scarps cut STEP volcanic chain. The preferred strikes of the major scarps are NE–SW and ENE–WSW, but minor N–S to N20°E striking scarps are also present, mainly in the eastern group (Fig. 2b). These scarps represent the morphological expression of faults with a prevailing dip-slip component although evidence of sinistral strike-slip movements has been recognized[26]. The spatial distribution of

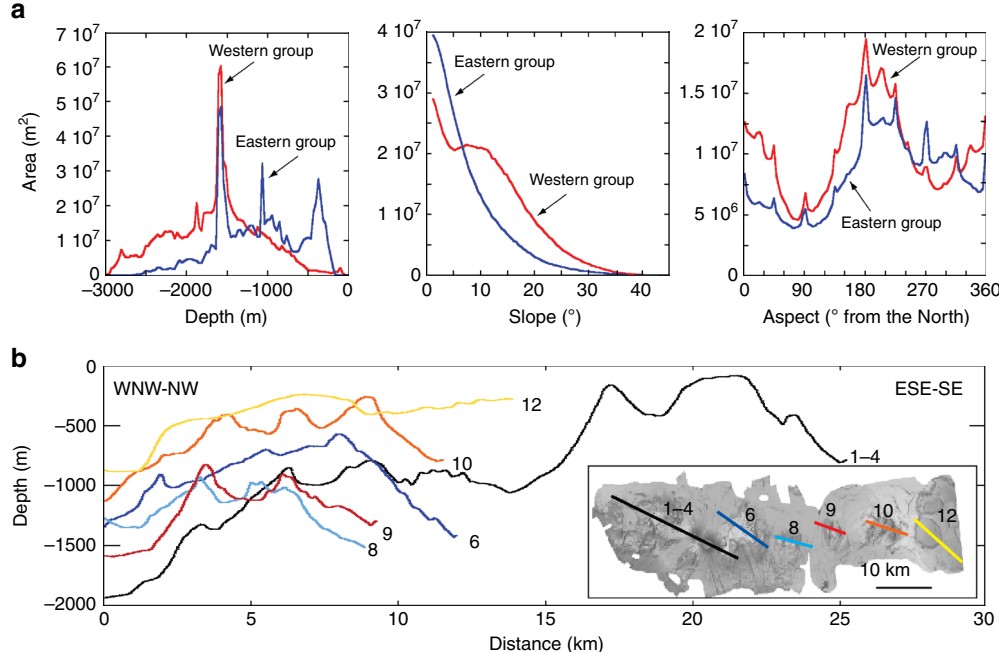

**Fig. 3** Morphometry of the Palinuro STEP volcanic chain. **a** Depth vs. area, slope vs. area, and aspect vs. area plots of the STEP volcanic chain as calculated from the morphometric parameters of the DTM in Fig. 2a (see auxiliary material). **b** Bathymetric profiles of the volcanoes 1–4, 6, 8, 9, 10, and 12 (location in **b**)

STEP volcanic chain vents show that most of them are concentrated in the western sector (Fig. 5a). This feature is mainly due to vents and the domes filling the caldera depression of the volcano 2 (Fig. 2b). The STEP volcanic chain western group is also characterized by a major N10°E vent alignment centered on the southern and northern flanks of the Palinuro Seamount and by a second-order E-W alignment centered on the volcano 8 (Fig. 5a). In the eastern group, N20°E and N-S alignments occur. A search of preferred orientations in the alignment of vents has been conducted by applying Fry's analysis[37] to the 80 STEP volcanic chain vents (Fig. 5b, left panel). The results are consistent with the above described structural arrangement. At a smaller spatial scale (12 km, Fig. 5b), the preferred alignment is E-W with minor relative maxima at ENE–WSW, NNE–SSW, and N-S. These strikes and those of the STEP volcanic chain faults and dikes mimic the theoretical orientation of the secondary shears associated to a major, sinistral wrench zone (Fig. 5b, right panel). At a larger scale, the distribution of the STEP volcanic chain vent alignments is flattened with a well-defined E-W preferred strike (24 and 48 km in Fig. 5b, left panel), which is that of the Palinuro STEP.

**Magnetic and gravity anomalies**. The STEP volcanic chain reduced to the pole (RTP) magnetic anomaly distribution shows a complex pattern (Fig. 6a). The westernmost flank of the Palinuro Seamount is dominated by low frequency and high amplitude negative values (up to −500 nT) related to the neighboring Marsili oceanic basin[38], where large negative magnetic anomalies of reversely magnetized oceanic crust partly mask the contribution of the Palinuro Seamount. High amplitude magnetic positive anomalies (up to 1700 nT) due to unaltered volcanic rocks characterize the flanks and the two main Palinuro cones, whereas two, less intense anomalies separated by a small E-W trending negative RTP anomaly characterize the westernmost STEP volcanic chain volcanoes (Fig. 6a). This magnetic pattern can be explained by the combination of highly magnetized fresh volcanic

rocks and hydrothermally altered products having lower to null magnetization[38, 39]. A well-known example of hydrothermally altered sector is represented by the caldera of volcano 2 (Fig. 2a, b), where hydrothermal and massive sulfides deposits have been recognized[39–41].

A weak amplitude RTP magnetic anomaly pattern characterizes the STEP volcanic chain eastern group. In this area, the RTP anomaly shows a semi-flat trend ranging from 0 to 100 nT without known evidence of hydrothermal alteration (Fig. 6a). This low magnetization can be related to a thin cover of the volcanic rocks, or to an upwelling of the Curie isotherm. Free air gravity anomaly of the STEP volcanic chain has been calculated by subtracting the latitude gravity contribution (normal gravity) from the gravity data using the parameters of the WGS84 model (Methods). A terrain correction allows us to take into account the variation in gravity signal generated by the steep gradient of the volcanoes. A complete Bouguer density reduction was estimated at 2.67 g/cm$^3$ and highlighting a clear NNE–SSW decreasing trend of the gravity signal, as expected moving from the high density oceanic crust of the Marsili basin to the lower density continental crust north of STEP volcanic chain (Fig. 6b). The Bouguer anomaly reduction provides an articulated distribution of local lows often centered on bathymetric highs, which results in a wider low-gravity pattern for the entire central and eastern sectors of STEP volcanic chain. Low values of the Bouguer gravity anomalies can be correlated to hydrothermally altered zones, as in the case of the caldera of the STEP volcanic chain western group[40].

**Modeling of magnetic and gravity data**. Quantitative analysis of the magnetic properties of STEP volcanic chain was first obtained by 2D inversion of the magnetic anomaly field (Fig. 7a). The recovered model provides information about the lateral distribution of rock magnetization estimated for a geometrically constrained crustal layer (Methods). The central sector of Palinuro Seamount records high values of magnetization distributed

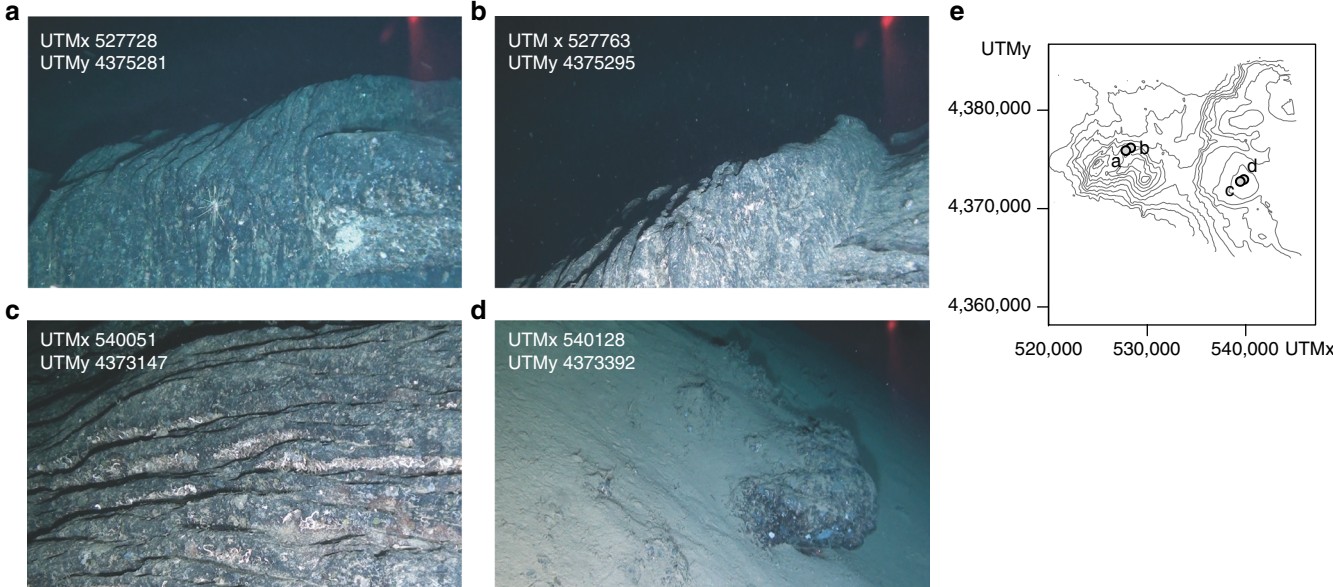

**Fig. 4** Images of the seafloor acquired with ROV. **a** Surface of a sheet lava flows. **b** Sub-vertical dike of the Enotrio Seamount. **c** Surface of a sheet lava flow and, **d** lava blob (pillow) of the flat-top edifice 12 (Fig. 2 for location). The UTM coordinates of the images and their map location (**e**) are also reported

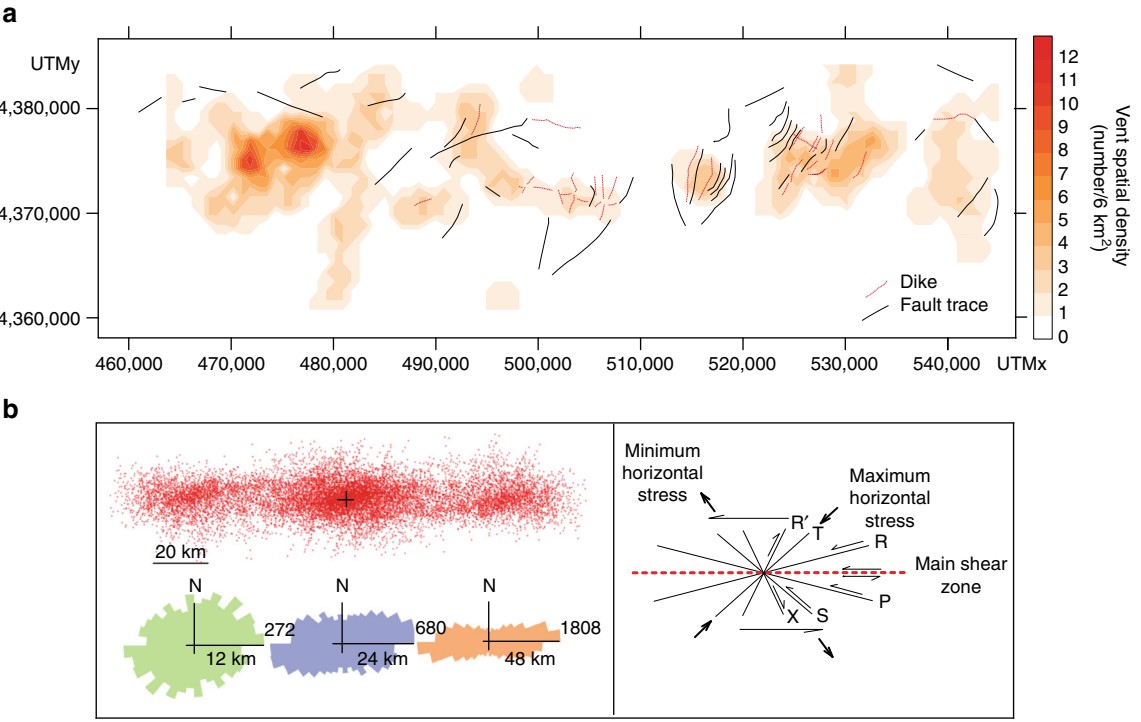

**Fig. 5** Spatial and structural relations between vents and faults. **a** Spatial density of the vents reported in Fig. 2b with the dikes and faults. **b** Results of the Fry's analysis[27] (red points) of the vents reported in Fig. 2b, and rose diagrams of the alignment of the points as a function of the distance from the center of the distribution; theoretical orientation and slip directions of R, R', P, X, and Y secondary shears relative to a main, sinistral shear zone according to ref. [43]. T is the extension fracture conjugate to R and R'

along the southern flank of the edifice and related to shallow intrusions and/or larger vent concentration (Figs. 5a, 6a). The STEP volcanic chain eastern group shows a low to almost null magnetization (Fig. 7a), which contrast with the recorded lava flow outcrops (Fig. 4). The low-magnetization pattern is confirmed by the results of a 3D inverse model (Fig. 7b; Methods), which shows conduit-like, sub-circular to slightly N-S elongated highly magnetized bodies centered below the volcanoes of the STEP volcanic chain western group, whereas significant magnetic structures cannot be recognized in the eastern group.

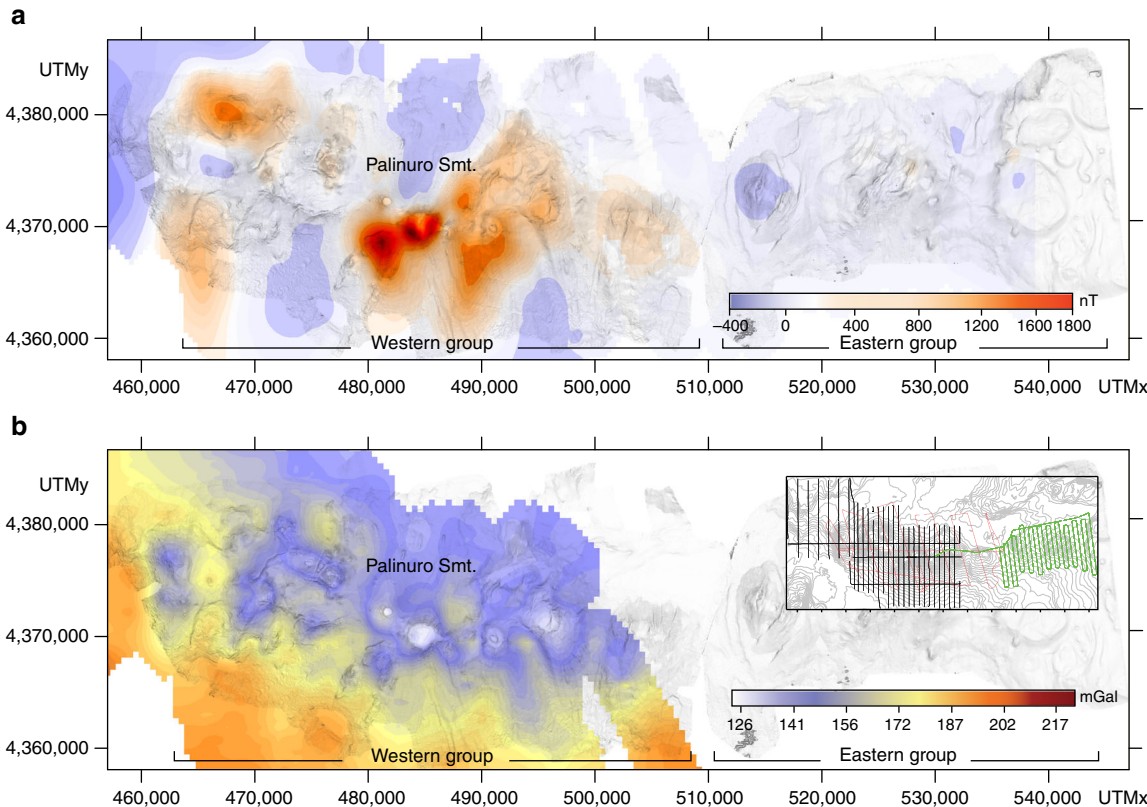

**Fig. 6** Magnetic and gravity anomaly maps. **a** RTP magnetic anomaly (grid cell size 400 m) overlaid on the shaded relief bathymetry of the STEP volcanic chain; **b** Bouguer gravity anomaly (grid cell size 500 m). Inset shows the survey lines: the gravity and magnetic survey acquired during PALI2008 cruise is reported in black, and the magnetic lines acquired during the Aeolian 2007 and SAFE 2015 cruises are reported in red and green, respectively

The highly magnetized bodies in Fig. 7b image the shallow plumbing system of the Palinuro Seamount and neighbor eruptive centers. Some of these bodies are roughly N-S elongated, thus suggesting a structural control on the magma intrusion pathways (Fig. 5). The poorly resolved magnetic structure of the STEP volcanic chain eastern group could be related to a reduced thickness of the lava flows or to a shallower depth of the Curie isotherm, as previously reported. To solve this ambiguity, we perform 2.5D joint gravity and magnetic forward models (Fig. 8; Methods). These models have been constructed by varying the geometry and properties (magnetization and density) of the seafloor rocks along and across a profile track (2.5D). The forward model technique accounts for the modification of the geometry of the causative bodies across the profile. We use the analytical signal approach, which combines the horizontal and vertical derivatives by clustering the solutions for dike-like sources (Structural Index SI = 0) along a 2.5D magnetic profile. We also computed the depth of the magnetic basement, which, in volcanic areas, could approximate the Curie isotherm (Methods). The results of the forward models are reported in two cross-sections (Fig. 8): a N-S striking section, constrained by seismic data across the Palinuro Seamount[42], and an E-W oriented section. In the N-S section, the gravity data are fitted by a thick crustal bedrock having the density value of flysh and carbonates (2400–2500 kg/m³), which are the two lithologies characterizing the basement of Palinuro Seamount[42] (Fig. 8a). The higher density and magnetized bodies are associated to volcanic rocks and show a maximum thickness of about 1.5 km (Fig. 8a). The low magnetic signature of the northern flank of the Palinuro Seamount is explained by the occurrence of hydrothermally altered deposits[30]. To the North, the volcanic products have a

limited extension. The depth of the magnetic basement follows the boundary between the volcanic rocks and the underlying continental crust, mainly below the Palinuro summit and southern flank (Fig. 8a). This depth abruptly increases from 1.5 to 4.5 km, according to a sudden thinning of the volcanic cover. Along the N-S profile, the depth of the magnetic basement marks the interface between the volcanic cover and the basement. The results of the models along the E-W section evidence highly magnetized volcanic rocks with different thickness covering the continental bedrock, with the exception of about a ~4 km wide central zone, where high density, low magnetization, and, probably, hydrothermally altered rocks occur (Fig. 8b). The depth of the magnetized basement is characterized by a set of highs and lows overlapping the interface between the volcanic rocks and the bedrock in the western part of the profile. In the eastern part, such overlap disappears and a relative high in the magnetic basement depth overlaps a zone of thicker volcanic rocks. In this part of the E-W profile, the magnetic basement lies at about 2 ± 0.4 km depth, a value that does not correlate with the thickness of the volcanic cover and bathymetry. This suggests a local upraising of the isotherms. In order to analyze the STEP volcanic chain eastern group, which shows a very-low-magnetic signature, we performed a 2.5D magnetic forward modeling focused on a single volcanic edifice. We select the Enotrio Seamount (Fig. 2), which is morphologically well raised with respect to the surrounding plain. Our modeling approach is based on two hypotheses. In the first case (layer A in Fig. 9), we suppose a volcanic cover with an average magnetization value of 5.5 A/m. The base of the volcanic covering is assumed to follow the slope of the seafloor surrounding the volcano. The recovered model (Fig. 9) provides a magnetic anomaly profile with values between −150 and 300 nT.

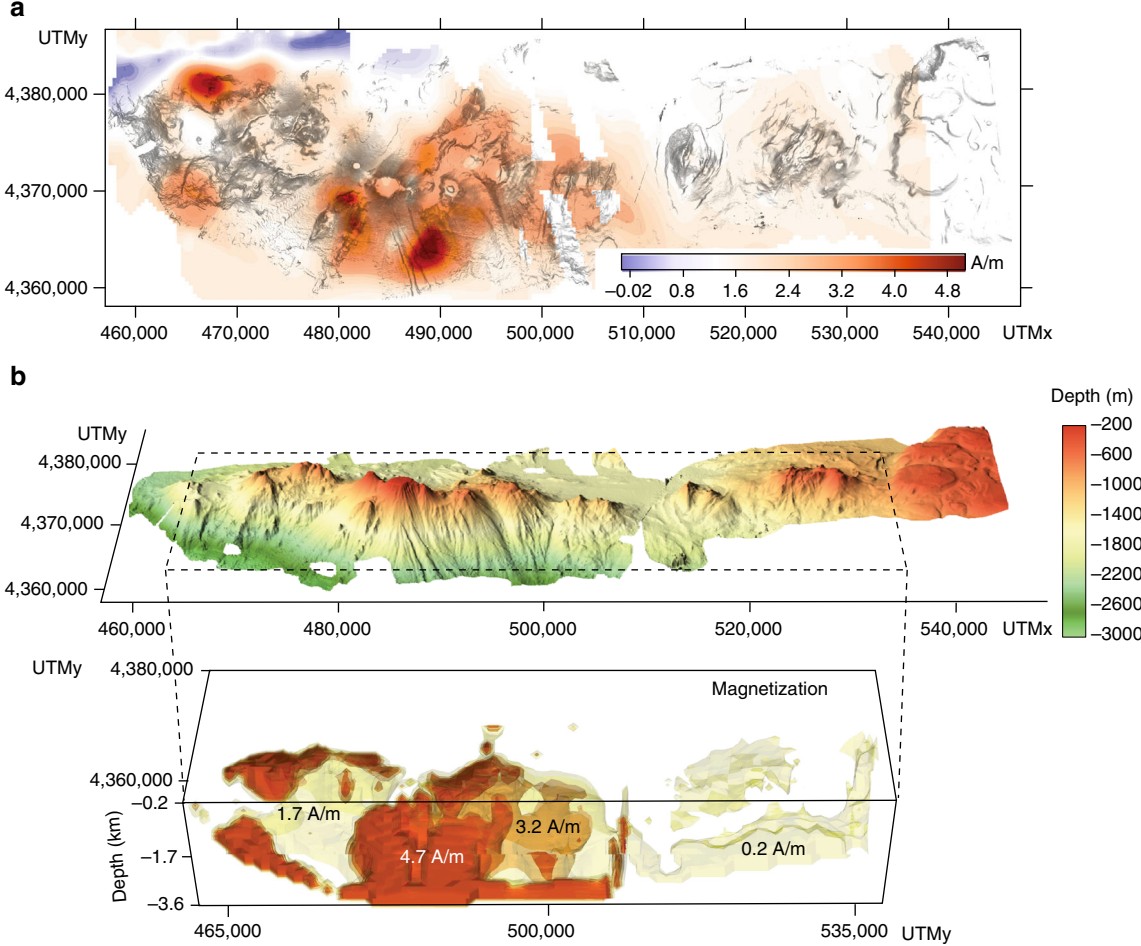

**Fig. 7** Results of the inverse models of the magnetic data. **a** Apparent magnetization intensity map derived from the 2D inversion of the magnetic data draped on the shaded relief bathymetry of the STEP volcanic chain. **b** 3D view of the STEP volcanic chain and results of the 3D inverse model, perspective view from the South

This model does not fit the observed data. In the second case (layer B in Fig. 9), we test the presence of a thermal anomaly. The good fitting was achieved by assuming a fully demagnetized bedrock due to high temperature. In this model, the volcanic products (5.5 A/m) have an average thickness of 120 m with peaks of 50 m. This is not geologically consistent with the elevation and thickness of the volcanic rocks of Enotrio Semount, but the model well reproduces the observed data (Fig. 9). From this result, we conclude that the low magnetic signature of the STEP volcanic chain eastern group can be explained by an upwelling of the Curie isotherm, which drastically reduces the thickness of magnetized crustal layer.

## Discussion

The fault arrangement of the STEP volcanic chain volcanoes and their spatial distribution and alignments (Figs. 2, 5) are consistent with those from experimental models of wrench tectonics[43, 44]. According to these models (Fig. 5b, right panel), the ENE–WSW striking faults may be interpreted as R (Riedel) shears of a major, E-W striking wrench zone while the NNE–SSW to NE–SW dikes dissecting the STEP volcanic chain eastern group volcanoes are consistent with R′ shears and T (tensional) cracks. At the scale of the whole STEP volcanic chain, the major volcanoes strike along the main shear zone of the Palinuro STEP, whereas, at the scale of the single edifice, the magma upraises along secondary shears.

This structural association defines an overall sinistral strike-slip zone moving, according to the focal mechanisms of earthquakes (Fig. 1), in response to a NE–SW striking maximum horizontal stress and a NW–SE striking minimum horizontal stress (Fig. 5b, right panel). The overall left-stepping WNW–ESE alignment of the volcanic edifices along the STEP may be explained in light of the results of experimental models of magma intrusion along strike-slip faults[45]. These models show that initially sub-circular intrusions emplaced along a sinistral strike-slip fault evolve toward elongated bodies as the shear deformation advances. The maximum elongation of these bodies follows the strike of P shears. This process is allowed by higher shear velocities along the fault and lower vertical uprising velocity of magma[45]. As a result, sub-circular magmatic intrusions aligned along a sinistral strike-slip fault will evolve toward left-stepping, en-echelon elongated bodies as the deformation progresses. This mechanism is able to explain the left-lateral arrangement of the major volcanoes along the STEP volcanic chain. A second mechanism could be the occurrence, in the basement, of pre-existing discontinuities reactivated as P shears during the strike-slip movements[46]. However, we exclude this latter mechanism because it is not supported by the available geological and geophysical data.

The morphology of the STEP volcanic chain western group volcanoes is comparable to that of Aeolian island-arc strato-volcanoes, which are central type, cone-like edifices with secondary vents located on the flanks and a summit crater or caldera

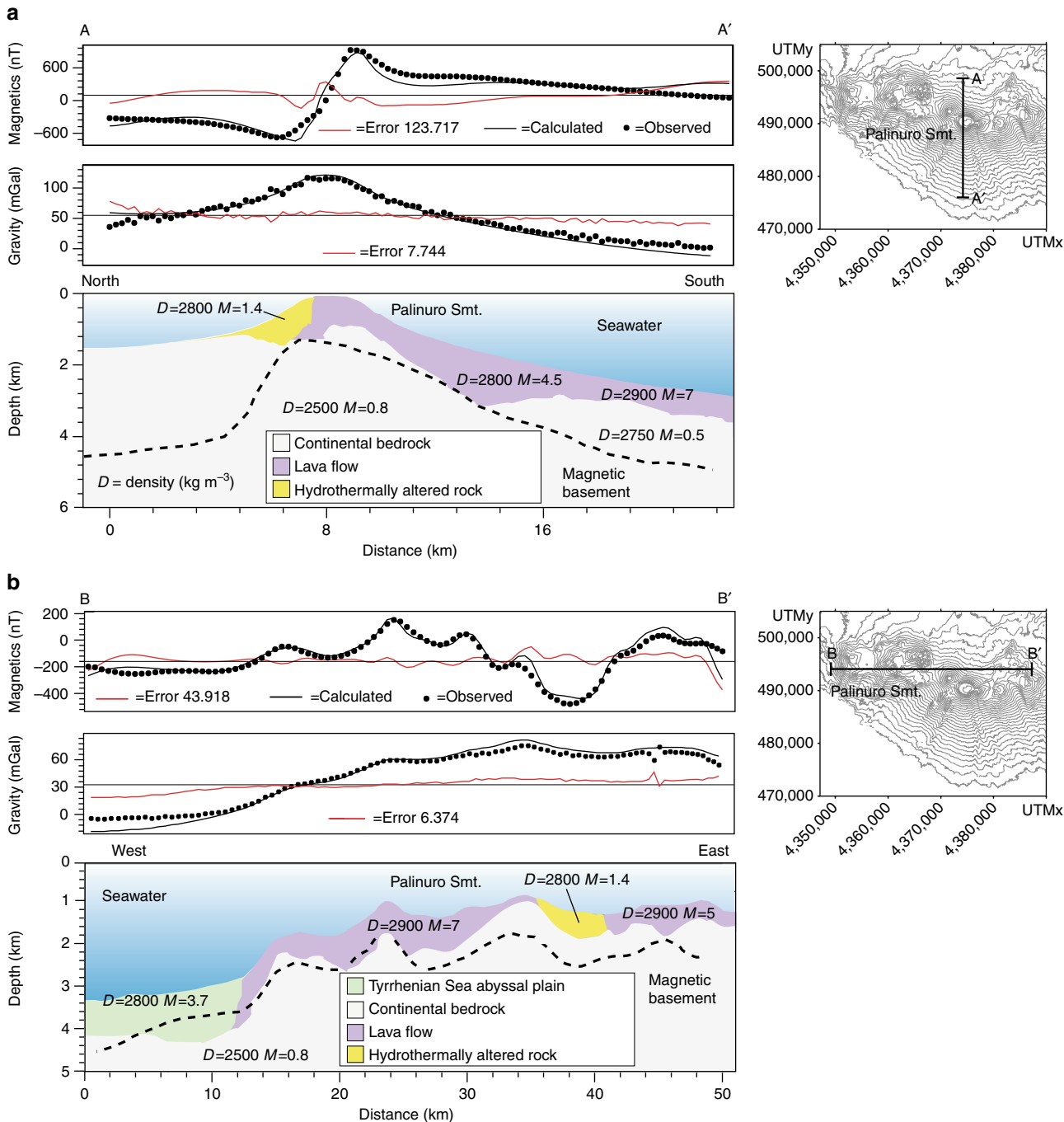

**Fig. 8** Results of the 2.5D magnetic and gravity forward models of the central sector of the Palinuro Seamount and surrounding volcanoes. **a** A–A′ is a N-S profile crossing the summit cone. **b** B–B′ is an E-W profile intercepting a caldera-like depression; the location of the profiles are reported in the insets. The dashed line represents the lower bound of the magnetized crustal portion by derivative computation. The dimensioning of the different bodies was estimated point by point as function of the effective distribution of seafloor features and are constrained by interpreted seismic lines[27, 42]

depression partly filled by domes or minor cones[19]. The shallow structure of the STEP volcanic chain western group is characterized by N-S elongated to sub-circular plumbing systems, thus implying the occurrence of magma reservoirs localized at depth. The occurrence of caldera depressions partly filled by domes (Fig. 2b), micro-earthquakes between 10 and 16 km depth below the Palinuro Seamount[29], and sill-shaped, high Vp/Vs anomalies at depth <20 km[21, 22] suggest that the reservoirs are stored in the middle–upper crust. The STEP volcanic chain eastern group show shapes similar to those recognized in

spreading centers and in the submerged portions of oceanic volcanic chains, thus reflecting prevailing fissural activity along dikes, localized spreading processes, and steady effusion rates. The results of the forward models suggest reduced thickness of the volcanic products consistent with a prevailing lateral growth of the volcanoes. The shallow (~2 km depth) Curie isotherm below the eastern group volcanoes inferred by the geophysical models suggests a general upwelling of the isotherms due to spreading processes. The strikes of the dikes affecting the STEP volcanic chain eastern edifices, which is NNE–SSW to NE–SW,

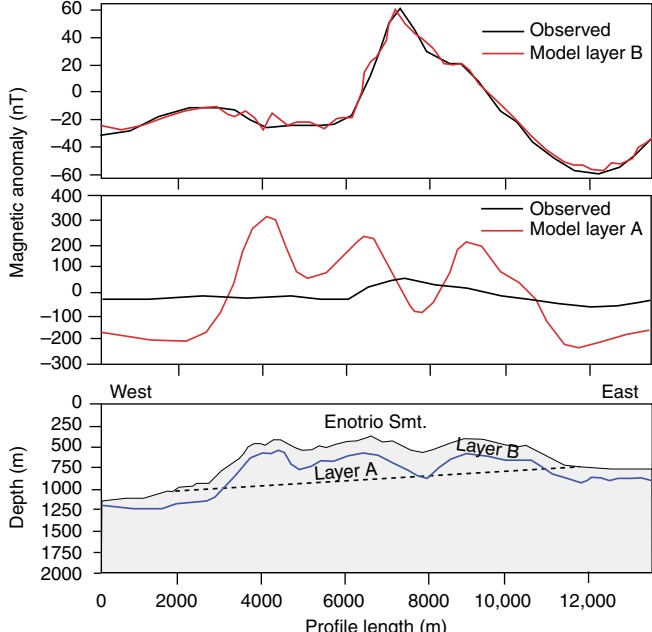

**Fig. 9** 2.5D magnetic forward models of the Enotrio volcano. Comparison between the recovered model and the observed magnetic anomaly assuming a distribution of high magnetized lava flows (5.5 A/m) confined within a layer B (blue continuous line in the bottom panel) and a distribution of the modeled magnetic anomaly profile with a distribution of high magnetized volcanic products (5.5 A/m) constrained between the bathymetry and an gently sloping plane (layer A, black broken line). The bedrock is supposed a continental crust with low magnetization (0.8 A/m)

suggest a dominant WNW–ESE to NW–SE direction of spreading. Such spreading acts a local scale and can be related to the second-order kinematics of the Palinuro STEP (Fig. 5b). Our data indicate that island-arc-type central stratovolcanoes and fissural vents coexist along the Palinuro STEP. To determine the volume of the STEP volcanic chain, we use a 0.002 arc resolution DTM of the Southern Tyrrhenian Sea (Methods). The calculated volume is compared with that of the Marsili ridge and of the Aeolian islands and seamounts. The results are summarized in Supplementary Table 3 and show that STEP volcanic chain has a volume of about 2700 km³, a value significantly larger than that of the Marsili ridge (856 km³). The volume of the Aeolian island-arc volcanoes including the associated seamounts is about 2550 km³. As a result, the size of the Palinuro STEP volcanoes is significantly larger than that of the neighbor back-arc spreading center and exceeds that of the island-arc edifices. This result may be explained by the complex pathways of the mantle melts in the Calabrian Arc-Ionian Sea subduction. Back-arc magmatism is generally related to the decompression melting of a mantle wedge possibly associated to the upraising of the asthenosphere[47], while the arc magmatism is mainly due to the partial melting of slab components and hydrated mantle[48]. According to this conceptual model, the available geophysical and geochemical data[14, 49, 50] of the Southern Tyrrhenian Sea area show an upraising of the asthenospheric mantle below the Marsili ridge and of lithospheric mantle and slab-derived fluids below the Aeolian island-arc (Fig. 10). The Palinuro STEP calc-alkaline volcanism is located at the northern boundary of the Ionian slab, where the above reported melting mechanisms may concurrently operate and where an additional toroidal mantle flow from the rear of the Ionian slab is confirmed by geophysical data[20, 25]. The trace element pattern of the Palinuro lava flows shows a IAB

signature[23, 50], but the isotopic composition of these lavas approaches that of the OIB-like Stromboli rocks[24]. Therefore, the Palinuro magmas originated from the interaction between an IAB-type metasomatized mantle source related to the melting of the slab[20, 24], and an OIB-type, asthenospheric source[24]. The OIB-mantle source is attributed to the inflow of asthenospheric mantle from the Africa foreland around the northern margin of the retreating slab[23, 24, 50], i.e., along the Palinuro STEP volcanoes. Our results show that central stratovolcanoes and fissural spreading centers coexist in subduction-transform edges. Such huge volcanism is strictly controlled by the second-order shears of the main wrench zone, and may be volumetrically comparable to that the island-arcs and larger than that of back-arc spreading ridges. We conclude that the STEP volcanoes are an important and, up to now underestimated, component of the subduction factory. Indeed, other subduction zones including the New Hebrides, Tonga, Lesser Antilles, and Sandwich are characterized by STEP along which submarine volcanoes align[51] (Supplementary Fig. 4). Therefore, the STEP volcanoes are a feature of many subduction zones. Most of these zones is characterized by a volcanic arc and back-arc spreading. The few subduction settings with non-volcanic STEP, as the Sulawesi (Supplementary Fig. 4) and Hellenic arcs, also lack of a well-developed volcanic arc and/ or back-arc. Our results and the above observations support the conclusion that the STEP volcanism requires the concurrent upraising of melts from different mantle sources including the wedge above the slab, the back-arc, and, through the STEP, the foreland mantle.

## Methods

**Data acquisition**. Since 2007, we collected high-resolution magnetics, gravity, and multibeam data during different oceanographic cruses (PALIN07, Aeolian_07, PALI08, Aeolian_2010; SAFE_2015). Geophysical surveys covered the major part of the volcanic system with a major emphasis for the northern and central sectors of the Palinuro volcanic chain, where hydrothermal manifestation were also identified[30, 31, 40, 41]. Data available on request from the authors.

**Bathymetry**. A first data set of depth measurements for the western sector of the Palinuro STEP volcanic chain was acquired during the Aeolian_2007 oceanographic cruise on board of the R/V Urania (January 2007). Data acquisition was carried out with a Reson Seabat 8160 multibeam sonar system. The apical part of the western sector was also investigated in the frame of the Aoelian_2010 cruise (May 2010), carried out on board of the R/V Urania by using a Simrad EM710-Kongsberg equipment. Finally, the eastern sector was acquired onboard of the Minerva Uno R/V (June 2016) with a Reson Seabat 7160 multibeam. All data were acquired by using a differential global positioning system, a motion sensor for heave, pitch, roll, and yaw real-time correction of depth measurements and a gyro. During each oceanographic cruise, speed of sound in the water column was regularly recorded and real-time applied at every 6–12 h. Data processing was carried out by using standard hydrographic software packages (i.e., Caris Hips and Sips, Fledermaus and PDS2000 swath model). Main processing steps were (1) quality check of beam forming and calibration checks; (2) removal of tide effects, (3) manual removal of spikes. The obtained data were re-organized in regularly spaced matrices (Digital Terrain Models or DTM) with an horizontal resolution varying from 2 m in shallow water (<200 m b.s.l.) to 25 m in deep water. All the data were merged in a final DTM (25 × 25 m), in agreement with larger values of grid cells given by deeper measurements. The final DTM covers about 1640 km² and it is geo-referenced in the UTM zone 33 WGS84. The acquisition details and the statistical parameters of the Palinuro DTM are reported in Supplementary Tables 1 and 2.

**Morphometric parameters**. The morphometry of the Palinuro STEP volcanic chain DTM has been conducted by calculating the following parameters: slope (in °), aspect (in ° from the North), general curvature, valley depth, and convergence index. The first three parameters and the convergence index have been calculated on a 3 × 3 moving window following the procedure detailed in ref. [52]. The valley depth has been determined following the method reported in ref. [53]. The convergence index has been used to construct the valley and ridge map. All the above DTM-extracted parameters maps are reported in Supplementary Fig. 1 of the auxiliary material.

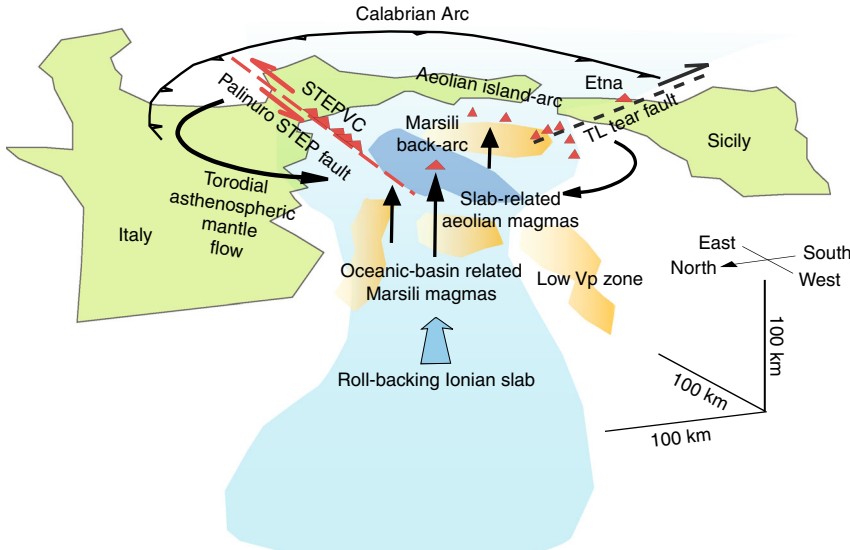

**Fig. 10** Conceptual geodynamic model of the Palinuro STEP volcanism. The location and extent of the STEP volcanic chain are from Fig. 2. The black arrows represent the zone of mantle upwelling from asthenospheric (Marsili back-arc, rear of the Ionian slab) or lithospheric (Aeolian Islands) sources according to geophysical[14, 15, 20-22] and geochemical data[50]. The toroidal mantle flow trajectories are from the geophysical data of ref.[25]. The velocity anomalies are identified according to refs.[14-23] and the Ionian slab is imaged on the basis of the earthquakes of Fig. 1

**Volume of volcanic edifices**. The volume calculation of the volcanic edifices of the Southern Tyrrhenian Sea and Aeolian Islands has been conducted by using a DTM with a 0.002 arc resolution provided by EMODnet Bathymetry Consortium 2016 EMODnet Digital Bathymetry (DTM), EMODnet Bathymetry (http://doi.org/10.12770/c7b53704-999d-4721-b1a3-04ec60c87238). We determine the volume of the Palinuro volcanic chain, Marsili Seamount, Aeolian Islands, and Aeolian seamounts (Supplementary Fig. 3). The boundary of each volcano has been delimited following ref.[54]. The volumes have been determined using the Surfer14 software by Golden Software. The calculated volumes are summarized in Supplementary Table 3. Errors in the volume calculations are <10%.

**Magnetics**. A large regional shipborne magnetic survey was carried out on board of R/V Aretusa (Italian Navy) during spring 2008 (Palin-2008 cruise). Data were collected using a line spacing distribution of 1 km apart to cover homogenously the central and western portion of Palinuro Seamount. Regional magnetic data of the central portion of Palinuro were also integrated using surveys by CNR during the Palin 2007 cruise on board of R/V Urania. During this survey, a large grid of N300° magnetic track lines were acquired crossing the caldera structure and the two shallow cones. Additional magnetic profiles were acquired during the Aeolian_07 oceanographic cruise on board of R/V Urania. The magnetic data set consists of 12 profiles oriented N150°E (5–6 km of average spacing between profiles). In addition, two magnetic profiles oriented N90°E were acquired for cross-check quality control of measurements, for a total amount of about 380 linear km. The easternmost portion of main edifice and Glabro volcanic structure were part of a detailed magnetic survey performed during SAFE 2015 cruise (RV Urania). This sector was mapped by using a set of N330 parallel lines spaced 1 km apart. The three data set shows different resolutions (line spacing) and distribution of the track lines (mean heading).

The three data sets have been merged into a unique database and then reprocessed, in order to get a clear magnetic anomaly grid of the Palinuro volcanic system. Each data set was corrected for spikes and outliers errors. Total intensity magnetic anomaly field was obtained by subtracting the IGRF model[55]. Cross-over errors were estimated and reduced by mean a statistical leveling procedure. Microleveling technique was also applied in order to remove high-frequency corrugation, mainly due to the not perfectly homogenous distribution of the line path.

Total magnetic anomaly field was than RTP by applying a FFT-phase shift transformation to correct for the skewness of the magnetic field at intermediate latitude. Average local values of inclination and declination of magnetic vector were derived from the IGRF model computed for the study area.

Processed magnetic anomaly data were thus inverted approaching a 2D inverse modeling for a quantitative estimation of the lateral distribution of apparent magnetization of the Palinuro and Glabro rocks. The inverse modeling was performed defining an uneven thickness mesh having as top boundary the bathymetry and a flat, oblique (southeastward dipping) plane as lower bottom. The inclination of the bottom layer was estimated considering the morphologic trend of Palinuro edifice, whose base is developed along a gently gradient from 1600 m at north to 3000 m at the south. Intensity of rock magnetization was estimated

sampling the magnetic anomaly by using a 400 m grid cell size and then inverting it by using 3D algorithm[56]. Optimal solution was achieved after 16 iterations with a RMS equal to 0.22.

**Gravity**. Ship borne gravity data were acquired during PALI20008 oceanographic cruise in concomitance of magnetic survey. Gravity data were collected using an Air/sea MicroG La Coste S136 gravity meter placed on board of R/V Aretusa. Raw readings were geo-referenced using GPS positioning with real-time differential correction (Marinestar gnss). Attitudes of the vessel (velocity, pitch, roll, and vertical acceleration) were provided by a Seapath Control Unit. On-board processing provided a roughly distribution of gravity data corrected for acceleration and Eotvos effect. A more accurate processing of data was carried out after data acquisition by performing a complete re-computation of the gravity data by starting from beam position and the cross coupling contribution (i.e., summation of all accelerations recorded by stabilized platform of the Air/sea system).

Relative gravity readings recorded by the Air/Sea meter were tied to an absolute gravity station (a node of Italian gravity network) located at the harbor of Naples (point of departure and arrive of the ship). Instrumental drift was also estimated and corrected using the gravity offset evaluated at the base station. The computed drift error was subtracted from the gravity data by using a linear trend. Smooth distribution of gravity field was achieved applying full leveling procedure based on cross-over matrix estimation. Leveled data acted as base for further reductions that allowed to identify local crustal density variation.

**3D inversion**. We derived a 3D model of the sub-seafloor by inverting the magnetic anomalies. The magnetic inversion is an interpretation technique aimed at deriving a subsurface 3D magnetization distribution reproducing the magnetic observations. Here, we have applied the inversion method described in ref.[56], where the mathematical algorithm is described in detail. The method is based on subdividing the sub-seafloor into a set of small prismatic cells to define complex magnetization distributions. Ambiguities inherent in the inversion process are reduced by introducing external constraints: (a) the magnetization direction is assumed to be parallel to the local geomagnetic field. For the Palinuro region, the magnetic field vector is characterized by an average inclination of 55.6° and declination 3°; (b) the magnetization has been, a priori bounded in the range 0–5 A/m based on expected magnetic signature for arc volcanoes-like type rocks and because the Palinuro is younger than the last geomagnetic polarity reversal; (c) a sharp, focused solution is found by minimizing the volume of the regions where the gradient of the magnetization distribution is significantly different from zero. These constraints select a specific optimal solution within the set of possible models fitting the observations within the same degree of accuracy and have proven particularly effective in the geophysical exploration of submarine volcanoes[56].

**Data availability**. All relevant data are available from the author Salvatore Passaro (salvatore.passaro@cnr.it).

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

## Acknowledgements

This study has been conducted with the funds from CNR and INGV.

## Author contributions

S.P. and G.V. conceived the study. S.P. collected and elaborated the bathymetric data and G.V. conducted the morphostructural analysis. L.C. elaborated the magnetic and gravity data, and the 2D and 2.5 models. F.C.T. contributed to the 3D magnetic model. All the authors wrote the text and contributed in the interpretation of the results.

## Additional information

**Competing interests:** The authors declare no competing financial interests.

