## [Peer Review File · Nature Communications]

Reviewers' comments:

Reviewer #1 (Remarks to the Author):

This manuscript describes a chain of submarine volcanic island in the Tyrrhenian Sea. The authors present bathymetric, magnetic and gravity data, and apply morphotectonic analysis, volume calculations of the volcanic edifices and 3D inversions. Altogether, they provide a compelling case for the existence of an ~E-W volcanic chain that has hitherto not be recognised. The possibility that volcanic activity was triggered by a sinistral slab tear fault makes a lot of sense and is supported by the preferred orientations of dykes and faults.

The idea that slab tearing can produce a chain of volcanoes is quite novel. It has been proposed earlier by Gvritzman and Nur (1999, Nature), Rosenbaum et al. (2008, Tectonics) and Gaspraon et al. (2009, J. Geodynamics), but the current contribution adds a lot more data to support this concept. The volume of volcanism is indeed impressive. In this respect, the impact of this contribution could be far more important than its contribution to understanding the geodynamics of the Tyrrhenian Sea, because the results imply that other volcanic edifices from the geological record may represent similar tear-related processes.

One thing that could further strengthen the authors' claims is an investigation of geochemical data, which could test whether or not magmatism was influenced by the process of tearing (e.g., through upwelling of mantle-derived melt). However, I don't know whether such data are available, and I think that the manuscript is sufficiently strong in its present form.

The manuscript is very well written and is quite succinct. The literature review is comprehensive with a reasonably fair assessment of previous literature. The methodological sections at the end of the manuscript are sufficiently detailed.

In conclusion, I think that this is an excellent manuscript, which is likely to attract a broad interest from the scientific community. I only have a few minor comments:

Line 72- is the acronym really needed? (I personally don't like abbreviations).

Line 75 - grammar

Line 77 - "been recognized"

Line 78 - it is a bit hard to see the symbol for pockmarks in fig 2b.

Line 126 - space missing "the contribution"

Line 147 - This is a very long paragraph. Perhaps consider subdividing it into 2-3 paragraphs.

Line 184 - grammar

Line 197 - space missing "by an"

Line 238 - "confirm this picture" is not the best expression

Line 330 - "on-board"

Line 374 -the reference seems to be incomplete

Gideon Rosenbaum

Reviewer #2 (Remarks to the Author):

This manuscript presents a new and very interesting bathymetric, potential-field data, and direct observations of the seafloor of the Palinuro seamounts. Based on the analysis of these new data, the authors show that these seamounts form a previously unrecognized submarine volcanic belt. The present analysis of the seafloor observations is very convincing and the proposed new understanding of the volcanic origin of the Palinuro seamounts is certainly very important for the tectonics of the Ionian Sea-Calabrian Arc.

I am much less convinced by the author's focus on the "Geodynamic significance of slab tear-fault volcanoes". I have, in particular, two comments about the author's geodynamical interpretation.

1. How reasonable is to generalize the geological features observed in the region of the Ionian Sea-Calabrian Arc to a more general understanding of the subduction? I would expect the authors to provide more information about similar geological settings (volcanoes built on to of tear faults in subduction zones) from other regions. How many places in the Earth's like this exists? How do they compare to the Palinuro seamount volcanoes in terms of tectonics, type of volcanism, and geodynamical settings?

2. In my view, the geodynamical interpretation itself (summarized in Fig. 6) is too subjective and lacks quantitative arguments. So far, the authors write that "the Palinuro STEP volcanoes are located in an area where back-arc astenospheric upraising, lateral mantle inflow from the foreland along STEPVC and arc-related lithospheric, hydrated mantle melts partly converge". What are possible quantitative arguments supporting this hypothesis?

I would suggest to clearly write in the abstract (and maybe in the title) that the paper describes the Palinuro seamounts.

Reviewer #3 (Remarks to the Author):

This study uses a multidisciplinary approach to define the tectono-magmatic features of the lateral edge of a back-arc basin, along a major left-lateral fault system in the southern Tyrrhenian Sea, Italy.

In particular, the Authors describe in detail and underline the important magmatic activity of this transform-like structure, pointing out that this may be a general feature accompanying the evolution of back-arcs in subduction zones.

The manuscript is in general clear, well written, sound and convincing, and deserves publication.

I would like to invite the Authors to better investigate and/or develop a first order feature, that is the overall left-stepping WNW-ESE alignment of the volcanic edifices along the STEP: this is quite obvious in the figures, but less obvious is its interpretation.

I wonder whether this may be related to the activity of left-stepping sinistral P shears, controlling the rise of magma in the areas of local extension created by their activity. This is a possibility which may be supported by any evidence of major WNW-ESE faults in the basement of the volcanoes.

However, in this case we should also explain why the P shears, and not the R shears, have been activated. In fact, one would expect that the R shears (WSW-ENE trending) are more active during the development of a strike-slip zone (as commonly observed), even though the volcanoes in this case are not aligned WSW-ENE.

In addition, I am sure that the manuscript would benefit from a very brief mentioning of other possibly similar cases in the world, to increase its impact and make it of interest to a broader audience. Therefore, I suggest that the Authors further develop the final part of their manuscript, briefly illustrating possible connections to other similar areas in the world.

Minor comments/questions/suggestions are directly annotated on the manuscript.

NCOMMS-17-10348 – answers to the questions/suggestions/comments by the reviewers.

Here below the questions (black color) and answers (blue color) to the comments and suggestions of the 3 reviewers. The sentences included in the new version of the main text are marked in *italic*.

We reorganize the headings of the different sections of the text according to the style of *Nature Communications* (Introduction, Results, Discussion, etc.).

Reviewer #1 (Remarks to the Author):

This manuscript describes a chain of submarine volcanic island in the Tyrrhenian Sea. The authors present bathymetric, magnetic and gravity data, and apply morphotectonic analysis, volume calculations of the volcanic edifices and 3D inversions. Altogether, they provide a compelling case for the existence of an ~E-W volcanic chain that has hitherto not be recognised. The possibility that volcanic activity was triggered by a sinistral slab tear fault makes a lot of sense and is supported by the preferred orientations of dykes and faults. The idea that slab tearing can produce a chain of volcanoes is quite novel. It has been proposed earlier by Gvritzman and Nur (1999, *Nature*), Rosenbaum et al. (2008, *Tectonics*) and Gaspraon et al. (2009, *J. Geodynamics*), but the current contribution adds a lot more data to support this concept. The volume of volcanism is indeed impressive. In this respect, the impact of this contribution could be far more important than its contribution to understanding the geodynamics of the Tyrrhenian Sea, because the results imply that other volcanic edifices from the geological record may represent similar tear-related processes.

One thing that could further strengthen the authors' claims is an investigation of geochemical data, which could test whether or not magmatism was influenced by the process of tearing (e.g., through upwelling of mantle-derived melt). However, I don't know whether such data are available, and I think that the manuscript is sufficiently strong in its present form.

Unfortunately, only 4 samples were dredged from the top of the Palinuro Seamount, and other samples from the 90 km long STEP volcanic chain are lacking. In addition, full isotopic data exist for only 2 sample. In the old version, at lines 240-244, we briefly reported the geochemical features of these two samples and their relations with the other geophysical data of the Southern Tyrrhenian Sea. In the new version, we add a fully referenced, more specific discussion on these geochemistry of these few samples and their significance within the geodynamic context of the Southern Tyrrhenian Sea and the mantle sources (lines 246-257): *The Palinuro STEP calc-alkaline volcanism is located at the northern boundary of the Ionian slab, where the above reported melting mechanisms may concurrently operate and where an additional toroidal mantle flow from the rear of the Ionian slab is confirmed by geophysical data^{20,25}. The trace element pattern of the Palinuro lava flows shows an IAB signature^{23,48}, but the isotopic composition of these lavas approaches that of the OIB-like Stromboli rocks²⁴. Therefore, the Palinuro magmas originated from the interaction between an IAB-type metasomatized mantle source related to the melting of the slab^{20,24}, and an OIB-type, asthenospheric source. The OIB-mantle source is attributed to the inflow of asthenospheric mantle from the Africa foreland around the northern margin of the retreating slab^{23,24,48}, i.e. along the Palinuro STEP volcanoes.*

The manuscript is very well written and is quite succinct. The literature review is comprehensive with a reasonably fair assessment of previous literature. The methodological sections at the end of the manuscript are sufficiently detailed.

In conclusion, I think that this is an excellent manuscript, which is likely to attract a broad interest from the scientific community. I only have a few minor comments:

Line 72- is the acronym really needed? (I personally don't like abbreviations).

The acronym has been deleted. STEPVC is changed in STEP Volcanic Chain

Line 75 – grammar

Corrected

Line 77 – “been recognized”

Corrected

Line 78 – it is a bit hard to see the symbol for pockmarks in fig 2b.

The symbol is now highlighted in blue and it is well evidenced in the figure.

Line 126 – space missing “the contribution”

Corrected

Line 147 - This is a very long paragraph. Perhaps consider subdividing it into 2-3 paragraphs.

The paragraph has been divided in three main paragraphs: an introductory one, a second one entitled *Forward magnetic and gravity models*, and a last paragraph entitled *Magnetic basement*.

The text has been not changed.

Line 184 – grammar

Corrected

Line 197 – space missing “by an”

Corrected

Line 238 – “confirm this picture” is not the best expression

We change in (now line 243): *According to this conceptual model, the available geophysical and geochemical data^{14,47,48} of the Southern Tyrrhenian Sea area show an upraising of the asthenospheric mantle below the Marsili ridge and of lithospheric mantle and slab-derived fluids below the Aeolian island-arc (Fig. 6).*

Line 330 – “on-board”

Corrected

Line 374 –the reference seems to be incomplete

Corrected

Reviewer #2 (Remarks to the Author):

This manuscript presents a new and very interesting bathymetric, potential-field data, and direct observations of the seafloor of the Palinuro seamounts. Based on the analysis of these new data, the authors show that these seamounts form a previously unrecognized submarine volcanic belt. The present analysis of the seafloor observations is very convincing and the proposed new understanding of the volcanic origin of the Palinuro seamounts is certainly very important for the tectonics of the Ionian Sea-Calabrian Arc.

I am much less convinced by the author's focus on the “Geodynamic significance of slab tear-fault volcanoes”. I have, in particular, two comments about the author's geodynamical interpretation.

1- How reasonable is to generalize the geological features observed in the region of the Ionian Sea-Calabrian Arc to a more general understanding of the subduction? I would expect the authors to provide more information about similar geological settings (volcanoes built on top of tear faults in subduction zones) from other regions. How many places in the Earth's like this exists? How do they compare to the Palinuro seamount volcanoes in terms of tectonics, type of volcanism, and geodynamical settings?

We add a new figure (Figure S4) in the Supplementary Information with the location of other STEP faults worldwide and the associated volcanic seamounts. In the text, we also add the following sentences

summarizing the relation between these STEP faults and the associated volcanoes. The new text is (268-276): *We conclude that the STEP volcanoes are an important and, up to now underestimated, component of the subduction factory. Indeed, other subduction zones including the New Hebrides, Tonga, Lesser Antilles and Sandwich are characterized by STEP faults along which submarine volcanoes align⁵⁶ (Fig. S4). Therefore, the STEP volcanoes are a feature of many subduction zones. Most of these zones is characterized by a volcanic arc and back-arc spreading. The few subduction settings with non-volcanic STEP, as the Sulawesi (Fig. S4) and Hellenic arcs, also lack of a well-developed volcanic arc and/or back-arc. Our results and the above observations support the conclusion that the STEP volcanism requires the concurrent upraising of melts from different mantle sources including the wedge above the slab, the back-arc, and, through the STEP faults, the foreland mantle.*

At the end of the Abstract we also add, as required in the annotated pdf, the following sentences: *The subduction-transform edge volcanism focuses localized spreading processes and its magnitude is underestimated. This volcanism characterizes the subduction settings associated to volcanic arcs and back-arc spreading centers.*

A full comparison between the volcanism of the Tyrrhenian STEP and that of the above cited similar geodynamic and tectonic settings cannot be done because, as in our case, the geochemical data are few and sparse (only few analysis on the rocks of seamounts are available; on average, 4 samples each 50 seamounts on <https://earthref.org/GERM/>, one of the more complete geochemical databases). Therefore, the above reported discussion, which is added in the new version, refers only to the geodynamic setting and tectonics (subduction settings with STEP, strike-slip fault(s)).

2. In my view, the geodynamical interpretation itself (summarized in Fig. 6) is too subjective and lacks quantitative arguments. So far, the authors write that “the Palinuro STEP volcanoes are located in an area where back-arc asthenospheric upraising, lateral mantle inflow from the foreland along STEPVC and arc-related lithospheric, hydrated mantle melts partly converge”. What are possible quantitative arguments supporting this hypothesis?

The sentence cited by the reviewer is not the our conclusion, it was based on the merging of our results and those from the cited literature. To avoid possible confusion in the reader (and in the referee), we delete this sentence. We now better present all the geochemical and geophysical evidences of ‘back-arc asthenospheric upraising, lateral mantle inflow from the foreland along STEP Volcanic Chain, and arc-related lithospheric, hydrated mantle melts’. These evidences come from previous studies, which were fully referenced also in the old text (ref. 14, 20,23, 47, 48, 49, 50). In detail, our new text is: *As a result, the size of the Palinuro STEP volcanoes is significantly larger than that of the neighbor back-arc spreading center and exceeds that of the island-arc edifices. This result may be explained by the complex pathways of the mantle melts in the Calabrian Arc-Ionian Sea subduction. Back-arc magmatism is generally related to the decompression melting of a mantle wedge possibly associated to the upraising of the asthenosphere⁴⁷, while the arc magmatism is mainly due to the partial melting of slab components and hydrated mantle⁴⁸. According to this conceptual model, the available geophysical and geochemical data^{14,49,50} of the Southern Tyrrhenian Sea area show an upraising of the asthenospheric mantle below the Marsili ridge and of lithospheric mantle and slab-derived fluids below the Aeolian island-arc (Fig. 6). The Palinuro STEP calc-alkaline volcanism is located at the northern boundary of the Ionian slab, where the above reported melting mechanisms may concurrently operate and where an additional toroidal mantle flow from the rear of the Ionian slab is confirmed by geophysical data^{20,25}.*

In addition, we briefly discuss the few available geochemical data as follows: *The trace element pattern of the Palinuro lava flows shows a IAB signature^{23,50}, but the isotopic composition of these lavas approaches that of the OIB-like Stromboli rocks²⁴. Therefore, the Palinuro magmas originated from the interaction between an IAB-type metasomatized mantle source related to the melting of the slab^{20,24}, and an OIB-type, asthenospheric source²⁴. The OIB-mantle source is attributed to the inflow of asthenospheric mantle from the Africa foreland around the northern margin of the retreating slab^{23,24,50}, i.e. along the Palinuro STEP volcanoes.*

We also change the legend of Fig. 6 by assigning to each symbol the correct reference to show that the proposed conceptual model bases on our and previously published geophysical and geochemical data. The new legend is: *Conceptual geodynamic model of the Palinuro STEP volcanism. The location and extent of the STEP Volcanic Chain are from Fig. 2. The black arrows represent the zone of mantle upwelling from asthenospheric (Marsili back-arc, rear of the Ionian slab) or lithospheric (Aeolian Islands) sources according to geophysical^{14,15,20-22} and geochemical data⁵⁰. The toroidal mantle flow trajectories are from the geophysical data of ref. 25. The velocity anomalies are identified according to ref. 14, 15, 20-23, and 25. The Ionian slab is imaged on the basis of the earthquakes of Fig. 1.*

I would suggest to clearly write in the abstract (and maybe in the title) that the paper describes the Palinuro seamounts.

We add the 'Palinuro volcanic chain' in the abstract, but not in title because, in light of the more general implications of our study, this may be misleading for the readers.

Reviewer #3 (Remarks to the Author):

This study uses a multidisciplinary approach to define the tectono-magmatic features of the lateral edge of a back-arc basin, along a major left-lateral fault system in the southern Tyrrhenian Sea, Italy. In particular, the Authors describe in detail and underline the important magmatic activity of this transform-like structure, pointing out that this may be a general feature accompanying the evolution of back-arcs in subduction zones. The manuscript is in general clear, well written, sound and convincing, and deserves publication.

I would like to invite the Authors to better investigate and/or develop a first order feature, that is the overall left-stepping WNW-ESE alignment of the volcanic edifices along the STEP: this is quite obvious in the figures, but less obvious is its interpretation. I wonder whether this may be related to the activity of left-stepping sinistral P shears, controlling the rise of magma in the areas of local extension created by their activity. This is a possibility which may be supported by any evidence of major WNW-ESE faults in the basement of the volcanoes. However, in this case we should also explain why the P shears, and not the R shears, have been activated. In fact, one would expect that the R shears (WSW-ENE trending) are more active during the development of a strike-slip zone (as commonly observed), even though the volcanoes in this case are not aligned WSW-ENE.

This is a very interesting point. In the new version we discuss this point by referencing to the results of experimental models on magma intrusions and ascent along strike-slip zones. The reference paper is: Corti G., Moratti G., Sani F., 2005. Relations between surface faulting and granite intrusions in analogue models of strike-slip deformation. *J. Struct. Geol.*, 27,1547-1562. These models clearly show that in a sinistral shear zone, an initially sub-circular intrusion evolve towards an elongated, elliptical body having the maximum

axis elongated along the strike of P shears. Therefore, magmatic bodies aligned along a main strike-slip fault will evolve towards en-echelon arranged elongated bodies as the deformation progresses. This mechanism is able to explain the left-lateral arrangement of the STEP volcanoes we observed. A second mechanism could be the occurrence, in the basement, of pre-existing discontinuities reactivated as P-shears during the strike-slip movements (Koyi, H. A., Ghasemi, A., Hessami, K. & Dietl, C. 2008. The mechanical relationship between strike-slip faults and salt diapirs in the Zagros fold–thrust belt. *Journal of the Geological Society, London* 165, 1031–44). However, this latter mechanism is not supported by the available geological/geophysical data.

In the new version of the text, we add the above reported references and discussed this point in the following way (lines 214-225): *The overall left-stepping WNW-ESE alignment of the volcanic edifices along the STEP may be explained in light of the results of experimental models of magma intrusion along strike-slip faults⁴⁵. These models show that initially sub-circular intrusions emplaced along a sinistral strike-slip fault evolve towards elongated bodies as the shear deformation advances. The maximum elongation of these bodies follows the strike of P shears. This process is allowed by higher shear velocities along the fault and lower vertical uprising velocity of magma⁴⁵. As a result, sub-circular magmatic intrusions aligned along a sinistral strike-slip fault will evolve towards left-stepping en-echelon elongated bodies as the deformation progresses. This mechanism is able to explain the left-lateral arrangement of the major volcanoes along the STEP Volcanic Chain. A second mechanism could be the occurrence, in the basement, of pre-existing discontinuities reactivated as P-shears during the strike-slip movements⁴⁶. However, we exclude this latter mechanism because it is not supported by the available geological and geophysical data.*

In addition, I am sure that the manuscript would benefit from a very brief mentioning of other possibly similar cases in the world, to increase its impact and make it of interest to a broader audience. Therefore, I suggest that the Authors further develop the final part of their manuscript, briefly illustrating possible connections to other similar areas in the world.

We add a new figure (Figure S4) in the Supplementary Information with the location of other STEP faults worldwide and the associated volcanic seamounts. In the text, we add the following sentences summarizing the relation between these STEP faults and the associated volcanoes. The new text is (268-276):

We conclude that the STEP volcanoes are an important and, up to now underestimated, component of the subduction factory. Indeed, other subduction zones including the New Hebrides, Tonga, Lesser Antilles and Sandwich are characterized by STEP faults along which submarine volcanoes align⁵⁶ (Fig. S4). Therefore, the STEP volcanoes are a feature of many subduction zones. Most of these zones is characterized by a volcanic arc and back-arc spreading. The few subduction settings with non-volcanic STEP, as the Sulawesi (Fig. S4) and Hellenic arcs, also lack of a well-developed volcanic arc and/or back-arc. Our results and the above observations support the conclusion that the STEP volcanism requires the concurrent uprising of melts from different mantle sources including the wedge above the slab, the back-arc, and, through the STEP faults, the foreland mantle.

At the end of the Abstract we also add, as required in the annotated pdf, the following sentences: *The subduction-transform edge volcanism focuses localized spreading processes and its magnitude is underestimated. This volcanism characterizes the subduction settings associated to volcanic arcs and back-arc spreading centers.*

Minor comments/questions/suggestions are directly annotated on the manuscript.

All the comments/questions of the annotated manuscript has been inserted in the new text with the exception of two questions: a) the reviewer suggest to use only STEP and not STEP fault(s) and b) he ask the significance of the term '2.5' referred to the gravity and magnetic modeling.

As concerns the question a), we change STEP instead of STEP faults only in places where it is more appropriate because the full definition of these structures is STEP faults (see the cited seminal paper on this structures: Govers et al. Lithosphere tearing at STEP faults: response to edges of subduction zones. Earth Planet. Sci. Lett. 236, 505–523 (2005)).

As concerns the question B, '2.5D' means that we can model the geometry of the causative bodies along a profile and also across a profile direction. This permits to obtained a more reliable fit of the data and lighten the modeling procedure. Simple 2D models (the modeling along a track) assume an infinite length of the body across the profile. This approach is mostly used for the interpretation of large scale structures like oceanic ridges. The applied method is in any case now better defined in the *Forward magnetic and gravity models* section, where we add the following sentences (lines 168-172): *These models have been constructed by varying the geometry and properties (magnetization and density) of the seafloor rocks along and across a profile track (2.5D). The forward model technique accounts for the modification of the geometry of the causative bodies across the profile. We use the analytical signal approach, which combines the horizontal and vertical derivatives by clustering the solutions for dike-like sources (Structural Index SI=0) along a 2.5D magnetic profile.*

Finally, the changes at figure 5 has been done (significant magnification of the text size).

REVIEWERS' COMMENTS:

Reviewer #2 (Remarks to the Author):

My comments were properly addressed and I consider that the manuscript could be accepted in its present form.